# Effect of an extension speech training program based on Chinese idioms in patients with post-stroke non-fluent aphasia: A randomized controlled trial

**Sun Pei[1], Li Weiwei[1], Zhang Mengqin[1], He Xiaojun**[1,2]*

**1** Department of Geriatrics, Renmin Hospital of Wuhan University, Wuhan, PR China, **2** Department of Cadre Health Office, Renmin Hospital of Wuhan University, Wuhan, PR China

* 13908653137@139.com

## Abstract

### Background

Chinese idioms have potential to act as preliminary training material in studies on post-stroke aphasia.

### Objective

To explore an extension speech training program that takes Chinese idioms as context and expands them into characters, words, sentences and paragraphs and evaluate the effects of this program in patients with post-stroke non-fluent aphasia.

### Methods

This was a randomized controlled trial. We recruited patients with post-stroke non-fluent aphasia from the Renmin Hospital of Wuhan University from January 2021 to January 2022. Participants were randomly assigned to group I and group II. Patients in group I had treatment with extension speech training based on Chinese idioms, and those in group II had treatment with conventional speech rehabilitation training. The training period in both groups was 40 min daily for 2 weeks.

### Results

A total of 70 patients (group I, n = 34; and group II, n = 36) completed the trial and were analyzed according to protocol. There were no significant differences in baseline values between both groups. After intervention, the scores of oral expression, comprehension, and reading in the Aphasia Battery Of Chinese scale and the scores of the Comprehensive Activities of Daily Living questionnaire significantly improved in both groups (P <0.05), with group I benefiting more (P <0.05).

**Data Availability Statement:** The data are available at the following link: DOI: 10.6084/m9.figshare.20766493; URL: https://figshare.com/articles/dataset/minimal_underlying_data_xlsx/20766493.

**Funding:** He Xiaojun, the author of the newsletter, received financial support.The fund comes from the project of Wuhan Science and Technology Bureau (Project No: 2013062301010820).http://kjj.wuhan.gov.cn. The funders had no role in study design, data collection and analysis, decision to publish, or preparation of the manuscript.

**Competing interests:** The authors have declared that no competing interests exist.

## Conclusion

This extension speech training program based on Chinese idioms can improve the language function and daily communication ability of the patients with post-stroke non-fluent aphasia.

## Trial registration

Chinese Clinical Trial Registry ChiCTR2000031825.

## Introduction

Post-stroke aphasia (PSA) is an acquired communication disorder caused by cerebrovascular disease damaging the language center of the dominant hemisphere of the brain. Patients with PSA usually have difficulty remembering words or lose the ability to speak, comprehend, read, or write. PSA is one of the most common and devastating symptoms of stroke, being present in 23%–40% of survivors [1, 2]. This communication barrier seriously impact on the patients' psychological [3], socials [4, 5], and daily life [6], causes a series of problems such as social isolation and unemployment [7, 8], and increases the mental and economic burden of caregivers [9]. The self-recovery of language function in patients with aphasia is very limited, and approximately 60% of them still have language impairment 12 months after onset [10]. How to actively carry out effective rehabilitation treatment and promote the recovery of patients' language function has attracted great attention in the field of post-stroke rehabilitation [11, 12].

Patients with non-fluent PSA usually have impaired language expression but retain their listening comprehension ability [13]. Language rehabilitation training is a clinical therapy supported by evidence-based medicine that can improve the speech function of patients with non-fluent aphasia [14]. Chinese idioms are four-character phrases with a fixed form and meaning that have been widely used for a long time. Chinese idioms are being used as training material in studies on aphasia rehabilitation and have demonstrated certain efficacy [15–17]. However, these studies have been limited to the repetitive reading of idioms themselves, which may not take full advantage of idioms and have limited the maximum effect of rehabilitation.

An important feature of Chinese idioms is that they have a complete and vivid story background. For example, "Yu Gong YI Shan," it tells the story of an old man named Yu Gong who finally moved a mountain through his constant efforts. This Chinese idiom teaches people that success is acquired only through enough efforts. Therefore, in the context of one idiom, relevant words, sentences, and paragraphs can be expanded, meaning that patients with aphasia can train in the same story background with different degrees of difficulty. This expandable rehabilitation material may be applicable to the whole cycle of rehabilitation for such patients.

Accordingly, this study aimed to explore an extension speech training program that takes Chinese idioms as context and expands them into characters, words, sentences and paragraphs and evaluate the feasibility and effectiveness of this extension speech training program based on Chinese idioms through randomized controlled trials. We hypothesize that the this program will be more effective at improving language function, as demonstrated by increased scores on the Aphasia Battery Of Chinese scale and the Comprehensive Activities of Daily Living questionnaire, than the conventional speech training program.

## Methods

### Design

We conducted a randomized controlled trial in the Renmin Hospital of Wuhan University from January 2021 to January 2022 to evaluate the effectiveness of speech training through

Chinese idioms. This study was reviewed and approved by the ethics committee of the Renmin Hospital of Wuhan University (approval number: WDRY2020-K229), and the trial was registered in the Chinese Clinical Trial Registry in April 2020 (registration number: ChiCTR2000031825).All data presented in the studies were deposited in an appropriate public repository (DOI: 10.6084/m9.figshare.20766493).

## Patients

We recruited patients with non-fluent PSA in the Department of Neurology, Neurosurgery and Rehabilitation of the Renmin Hospital of Wuhan University from January 2021 to January 2022. Patients who met the inclusion and exclusion criteria, agreed to participate in the study, and signed an informed consent form were included in the study. The following criteria were used:

**Inclusion criteria.**

a. First stroke patients with a definite clinical diagnosis by brain CT scan or MRI examination.

b. Non-fluent aphasia diagnosed by the aphasia screening scale.

c. Stable condition, clear consciousness, and no cognitive impairment.

d. Native language is Chinese.

e. Age >18 years.

f. An education level of primary school and above.

**Exclusion criteria.**

a. Suffering from other serious diseases or intolerable examination.

b. Complete aphasia, wernicke aphasia, transcortical sensory aphasia, and transcortical mixed aphasia with severe hearing comprehension impairment.

c. Visual defect, hearing impairment, dysarthria, and speech apraxia.

d. Previous mental history.

## Sample size

The software Gpower (HHU, Germany, Dusseldorf)was used to calculate the sample size [18]. An effect size (Cohen's d)of 0.8, α (alpha) value of 0.05, and statistical power of 0.8 were required. Two independent samples were selected for the t-test. Considering that 15% of samples were withdrawn, 62 cases (31 cases in the group I and 31 cases in the group II) were required to be included in the study.

## Study groups

After the participants signed the consent form, we randomly assigned them into group I (Chinese idioms training) and group II (routine rehabilitation training) in a 1:1 manner. We used the random envelope method for grouping concealment. After preparing the randomization plan, the sequentially-coded, opaque, and sealed envelope was used. Each grouping plan was put into an opaque envelope, the code was written outside of it, and the envelope was sealed and given to the researcher. After each participant entered the study, the researcher will

determine the eligibility of the subject, subsequently open the envelope with the corresponding number, and intervene accordingly. As this study involved changes in the content and form of rehabilitation training, it was impossible to blind participants and research intervention personnel, yet blinding will be used on data collection and analysis personnel to avoid bias to a great extent.

## Intervention

**Group I.** Group I was given routine nursing and drug treatment, as well as extension speech training based on Chinese idioms as follows:

**Intervention materials:** A total of 50 familiar Chinese idioms were collected in this study. The researchers and experts of the Chinese Department worked together to unpack or expand idioms into characters, words, sentences, and paragraphs according to the story background of each idiom and make an idiom training book. Every idiom and its expanded form will be accompanied by pictures to stimulate imagination and improve interest. Taking the idiom "Yu Gong Yi Shan " as an example, the training contents of characters, words, sentences, and paragraphs is illustrated in S1 Fig.

**Rehabilitation training:** First, the language function of patients before each training was evaluated. Patients with poor language function were guided start from the reading training of characters and words, whereas those with better language function were allowed to practice oral expression at the sentence or paragraph level. The rehabilitation therapist first shows the corresponding pictures of idioms to the patients and subsequently guides them into reading the text or asks them relevant questions according to the idiom training manual. The therapist was to encourage the patient to speak as much as possible, giving affirmation whenever the patient reads aloud or answers correctly. When the patients' oral expression is unclear, they were timely corrected.

**Period of intervention:** The training period was 40 min daily for 2 weeks and was able to be divided according to the patient's state.

**Group II.** The control group was given routine nursing and drug treatment, as well as routing speech rehabilitation training as follows:

**Respiratory regulation:** The patients were guided to inhale deeply through the nose, shrink lips and exhale, and practice nasal, pharyngeal, and oral atresia.

**Muscle exercise:** Patients were instructed through tongue muscle and facial muscle training and were guided to practice mumps, tongue rolling, chewing, tongue swing left and right, forward extension, and backward contraction.

**Pronunciation practice:** The instructor was to demonstrate the shape of the mouth and explain the position of pronunciation organs. The patient performs vowel pronunciation practice, including "a", "o", "e", "u", and "ü".

**Period of intervention:** As in Group I, the training period was 40 min daily for 2 weeks and was able to be divided according to the patient's state.

## Assessment of treatment outcomes

The Aphasia Battery of Chinese (ABC) scale is the main efficacy index to evaluate the language ability of participants. The ABC scale is compiled by combining the internationally recognized Boston Diagnostic Aphasia Examination (BDAE) and Western Aphasia Battery (WAB). It has good reliability and validity [19], and the content of the scale includes four sections: oral expression, comprehension, reading and writing. The total score of each section is 100. The higher the score, the better the language function of this section.

The Comprehensive Activities of Daily Living (CADL) scale was used as the secondary index to evaluate the language ability of participants. The examination mainly includes 22 items, in addition to 34 sub-items, of daily life communication activities. It aims to obtain objective results by assessing the daily life communication ability of patients with aphasia and guide language training. The total score of the scale is 136. The higher the score, the better the communication ability [20].

Baseline characteristics were collected for each patient, including age, sex, education, stroke type, aphasia type, and hemiplegia side. Prior to intervention, the therapists assessed participants using the ABC Scale and CADL questionnaires and continued follow-up at the end of the treatment period. Data were analyzed by a intent-to-treat analysis and the missing data were filled out using the last-observation-carried-forward.

## Statistical analyses

SPSS version 26.0 (IBM, Armonk, NY) was used to process and analyze the data. Normally distributed data are expressed by mean (SD) and non-normally distributed data are expressed by Median(interquartile range, IQR). For the comparison of paired measurement data, the paired sample t-test was used for normally distributed data, and the Wilcoxon signed rank test was used for non-normally distributed data. As for the comparison of two-sample measurement data, the two independent sample t-test was used for normally distributed data, whereas the two independent sample Wilcoxon signed rank test was used for non-normally distributed data. The Chi-square test was used to classify data. A p-value $< 0.05$ was considered statistically significant.

## Results

From January 2021 to January 2022, a total of 124 patients were recruited, and 70 patients (group I, n = 34; and group II, n = 36) met the criteria and agreed to participate in this study. Three participants (group I, n = 1; and group II, n = 2) withdrew halfway during the intervention process, and all participants were finally analyzed. **Fig 1** shows the study flow diagram according to the Consolidated Standards of Reporting of Trials guidelines.

A total of 20 women (29%) and 50 men (71%) were included in this study, with an average of 65.0 (11.3) years. The educational background mainly comprised junior middle school (n = 29, 41.4%) and senior high school (n = 22, 31.4%). The main types of stroke were ischemic stroke (n = 56, 80.0%), Broca's aphasia (n = 41; 57.1%), and Transcortical motor aphasia (n = 29; 41.4%). Most patients had right hemiplegia (n = 39, 55.7%), and the median of intervention period was 15(1) days. All patients participated in the expression, comprehension, and reading of ABC and scoring of all CADL items. Moreover, 17 patients without hemiplegia and 11 patients with left hemiplegia participated in the scoring of ABC writing items (group I, n = 13; and group II, n = 15). Baseline data are shown in **Table 1**. There is no significant difference between the intervention and control groups in age, gender, education level, stroke type, aphasia type, hemiplegia side, and intervention time.

## Language function evaluation

All participants who completed intervention were evaluated for oral expression, comprehension and reading through the ABC scale. Moreover, 28 patients (group I, n = 13; and group II, n = 15) participated in the writing evaluation. There is no significant difference between the intervention and control groups in initial ABC score(all P > 0.05).The scores of oral expression, comprehension, reading were significantly higher following intervention than before intervention in both groups (all P <0.05), and group I underwent a better effect (all P <0.05).

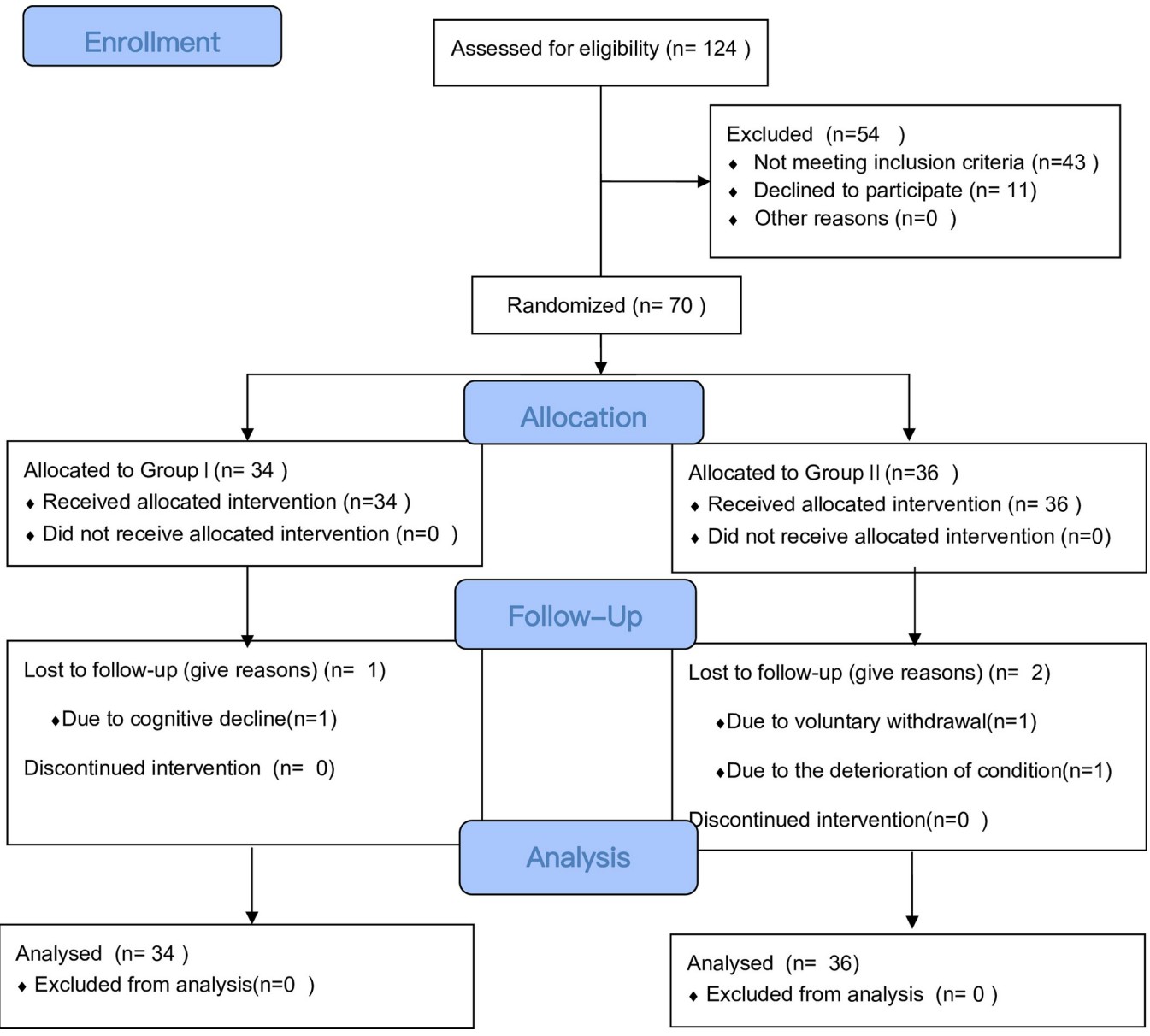

**Fig 1. Flow diagram.**

Nevertheless there was no significant difference in the scores of writing before and after the intervention in both groups(both P > 0.05) (**Table 2**).

## Daily communication skills

The daily communication ability of patients with aphasia was evaluated using the CADL scale. There is no significant difference between the intervention and control groups in initial CADL scores (P > 0.05). The CADL scores of both groups significantly improved following intervention compared to values before intervention (both P <0.05), with group I demonstrating more of an improvement than group II (P <0.05) (**Table 3**).

**Table 1. Comparison of the two groups' general clinical data.**

| | | Group I n = 34 | Group II n = 36 | P-value |
|---|---|---|---|---|
| Gender (n) | | | | 0.705 |
| | Female | 9 | 11 | |
| | Male | 25 | 25 | |
| Age(years) (mean (SD)) | | 65(10.7) | 64.7(13.3) | 0.916 |
| Education level (n) | | | | 0.807 |
| | Primary school | 4 | 7 | |
| | Junior high school | 14 | 15 | |
| | High school | 12 | 10 | |
| | University degree or above | 4 | 4 | |
| Stroke type (n) | | | | 0.632 |
| | infarction | 28 | 28 | |
| | hemorrhage | 6 | 8 | |
| Aphasia type (n) | | | | 0.598 |
| | Broca's aphasia | 21 | 20 | |
| | Transcortical motor aphasia | 13 | 16 | |
| Hemiplegia side (n) | | | | 0.929 |
| | Left | 5 | 6 | |
| | Right | 20 | 19 | |
| | Bilateral | 1 | 2 | |
| | None | 8 | 9 | |
| Intervention time(days) (Median(IQR)) | | 15(1) | 14(1) | 0.783 |

Differences between groups in classify data(gender, education level, stroke type, aphasia type and hemiplegia side), age and intervention time were analysed by Chi-square test, two independent sample t-test and Wilcoxon signed rank test respectively.

## Discussion

In the present study, we found that after a 2-week intervention, both group I (Chinese idioms training) and group II (routine rehabilitation training) showed significant improvement in the scores of oral expression, comprehension and reading sections in ABC and the scores of CADL, and the improvement in group I was significantly better than that in group II(The average improvement rates of oral expression, comprehension, reading and CADL were 48%, 13%, 45.2% and 55.9% in group I and 36.4%, 6.3%, 18.1% and 33.3% in group II, respectively). This indicates that the rehabilitation training based on Chinese idioms can effectively improve the language function and daily communication ability of patients, and its effect is better than that of conventional rehabilitation methods. Simultaneously, we also found that patients can benefit from the rehabilitation training of Chinese idioms in listening comprehension, even if they retain better in this ability.

The benefit of the patient's language function in this program may be related to the characteristics of the training material. Chinese idioms are harmonious, with level and oblique deployment and a strong sense of rhythm [21, 22]. This rhythm of idioms can be used to prepare for the rapid start of pronunciation by adjusting the speed of pronunciation more clearly similar to Melodic Intonation Therapy [23]. Moreover, the structure of idioms is fixed, and the corresponding collocations are neat. Idioms are composed of four characters, and there is a high correlation between the first and second character halves [24]. Idioms, as a formalized way of language expression, can be activated earlier and higher in the process of speech

**Table 2. Comparison of the ABC scores before and after the intervention in both groups.**

| | | Intent-to-treat Analysis | | |
|---|---|---|---|---|
| | | Group I | Group II | P-value |
| | | n = 34 | n = 36 | |
| ABC score (Mean (SD)) | | | | |
| Expressing | Before the intervention | 55.8(21.7) | 52.4(25.9) | 0.555 |
| | After the intervention | 82.6(13.3) | 71.5(22.6) | 0.016* |
| | P-value | <0.001* | <0.001* | |
| Comprehension | Before the intervention | 77.8(7.3) | 75.9(10.2) | 0.395 |
| | After the intervention | 87.9(9.6) | 80.7(11.5) | 0.006* |
| | P-value | <0.001* | <0.001* | |
| Reading | Before the intervention | 55.8(19.0) | 57(24.2) | 0.822 |
| | After the intervention | 81(15) | 67.3(23.78) | 0.006* |
| | P-value | <0.001* | <0.001* | |
| Writing | Before the intervention | 78(13.7) | 75.9(22.2) | 0.881 |
| | After the intervention | 81.9(15.4) | 80.7(18.7) | 0.858 |
| | P-value | 0.051 | 0.123 | |

Analysed by the two independent sample t-test between groups and paired sample t-test within groups.

*P <0.05

generation and subsequent extraction and processing [25] and stimulate the potential of language processing in the right hemisphere of the brain [26], thus promoting the ability of language extraction and expression in patients [27].Currently, there are a few studies on rehabilitation training through Chinese idioms. Li 's research found that idiom reading training can improve the oral output ability of patients with stroke [17], and Zhou's research found that idiom reading training can improve the speech rehabilitation and self-efficacy ability of post-stroke patients with non-fluency aphasia [15]. These research results are consistent with the results of our study.

The coherence of training material context may be the influencing factor of oral expression in aphasia. Studies have shown that providing contextual sentences helps patients with aphasia better understand [28]. Based on the training of reading idioms out loud, this study has expanded such a training and developed an extension speech training program that integrates characters, words, sentences, and paragraphs. This improved idiom training program can provide patients with aphasia training in the same context.

We recruited patients with non-fluent PSA, who are the main population of clinical rehabilitation training due to their retention of understanding ability, correct semantic expression,

**Table 3. Comparison of the CADL scores before and after the intervention in both groups.**

| | | Intent-to-treat Analysis | | |
|---|---|---|---|---|
| | | Group I n = 34 | Group II n = 36 | P-value |
| CADL score (Mean (SD)) | | | | |
| | Before the intervention | 70.8(33.3) | 70.7(38.4) | 0.986 |
| | After the intervention | 110.4(25) | 94.3(33.3) | 0.026* |
| | P-value | <0.001* | <0.001* | |

Analysed by the two independent sample t-test between groups and paired sample t-test within groups.

*P <0.05

and high degree of cooperation [29]. Based on the results of randomized controlled trials, we preliminarily found that idiom rehabilitation training may be beneficial to the rehabilitation for non-fluent PSA. Nevertheless, this study has certain limitations. First, although the sample size meets the needs of this trial, it is too small and limited to the same hospital. Second, the intervention period of our study is relatively short, and the long-term effects are not monitored. Third, despite the training of data collectors, the evaluation results of the scale may still be subjectively affected, and the research results lack an objective evaluation index. Lastly, as this study is using Chinese idioms, these results could not be generalized more broadly for an international audience directly. However, each language has a large repertoire of idiomatic expressions; we hope our research may have implications for the use of idiomatic expressions in other languages.

In conclusion, this randomized controlled trial shows that idiom rehabilitation training can improve the language function and daily communication ability of patients with non-fluent PSA, which is feasible and effective.

## Supporting information

**S1 Checklist. CONSORT 2010 checklist.**
(DOC)

**S2 Checklist. PLOSOne clinical studies checklist.**
(DOC)

**S1 Protocol. Study protocol.**
(PDF)

**S2 Protocol. Translation of study protocol.**
(PDF)

**S3 Protocol. Ethics approval.**
(PDF)

**S4 Protocol. Translation of ethics approval.**
(PDF)

**S1 Fig. Examples of idiom materials.**
(PDF)

## Acknowledgments

We thank the Department of Neurology, Neurosurgery and Rehabilitation of the Renmin Hospital of Wuhan University for providing a platform for the development of this study. We would like to thank the speech therapist Ms. Yang HM for providing relevant training for this study. We would like to thank the neurologist Professor Li T and the rehabilitation doctor Professor Zhu SS for their technical support. Finally, we would like to sincerely thank the patients with post-stroke aphasia who participated in this study.

## Author Contributions

**Conceptualization:** Sun Pei, He Xiaojun.

**Data curation:** Sun Pei, Li Weiwei.

**Formal analysis:** Sun Pei, Li Weiwei, Zhang Mengqin.

**Funding acquisition:** He Xiaojun.

**Investigation:** Sun Pei, Li Weiwei.

**Writing – original draft:** Sun Pei, Li Weiwei, Zhang Mengqin, He Xiaojun.

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
