## [Decision Letter · Decision Letter 0]

23 Aug 2022

PONE-D-22-14223The effect of an extension speech training program based on Chinese idioms in post-stroke nonfluent aphasia:a randomized controlled trialPLOS ONE

Dear Dr. Xiaojun

Thank you for submitting your manuscript to PLOS ONE. After careful consideration, we feel that it has merit but does not fully meet PLOS ONE’s publication criteria as it currently stands. Therefore, we invite you to submit a revised version of the manuscript that addresses the points raised during the review process.

 The authors should  follow the instruction from the reviewer. Please revise the method on random allocation and method of assessment. 

Please submit your revised manuscript by  Oct 07 2022 11:59PM. If you will need more time than this to complete your revisions, please reply to this message or contact the journal office at plosone@plos.org. Please include the following items when submitting your revised manuscript:A rebuttal letter that responds to each point raised by the academic editor and reviewer(s). You should upload this letter as a separate file labeled 'Response to Reviewers'.A marked-up copy of your manuscript that highlights changes made to the original version. You should upload this as a separate file labeled 'Revised Manuscript with Track Changes'.An unmarked version of your revised paper without tracked changes. You should upload this as a separate file labeled 'Manuscript'.

We look forward to receiving your revised manuscript.

Kind regards,

Rizaldy Taslim Pinzon

Academic Editor

PLOS ONE

Journal Requirements:

2. PLOS requires an ORCID iD for the corresponding author in Editorial Manager on papers submitted after December 6th, 2016. Please ensure that you have an ORCID iD and that it is validated in Editorial Manager. To do this, go to ‘Update my Information’ (in the upper left-hand corner of the main menu), and click on the Fetch/Validate link next to the ORCID field. This will take you to the ORCID site and allow you to create a new iD or authenticate a pre-existing iD in Editorial Manager. Please see the following video for instructions on linking an ORCID iD to your Editorial Manager account: https://www.youtube.com/watch?v=_xcclfuvtxQ.

“He Xiaojun, the author of the newsletter, received financial support.The fund comes from the project of Wuhan Science and Technology Bureau (Project No: 2013062301010820).http://kjj.wuhan.gov.cn”

6. We note that the original protocol that you have uploaded as a Supporting Information file contains an institutional logo. As this logo is likely copyrighted, we ask that you please remove it from this file and upload an updated version upon resubmission.

Reviewers' comments:

Reviewer's Responses to Questions

**Comments to the Author**

1. Is the manuscript technically sound, and do the data support the conclusions?

Reviewer #1: Partly

2. Has the statistical analysis been performed appropriately and rigorously? 

Reviewer #1: I Don't Know

3. Have the authors made all data underlying the findings in their manuscript fully available?

Reviewer #1: Yes

4. Is the manuscript presented in an intelligible fashion and written in standard English?

Reviewer #1: No

5. Review Comments to the Author

Reviewer #1: This study was a randomized controlled trial in which patients with post-stroke non-fluent aphasia were randomly assigned to either treatment with speech training based on Chinese idioms or treatment with conventional speech rehabilitation. The study included 67 patients and took place over a two-week period. The intervention is an interesting one and could likely be generalized to other countries. The authors could greatly improve the paper by providing more details and the results than are given. I am also not sure why they would have conducted a per-protocol, rather than an intent-to-treat, analysis. What was their reasoning for this?

Other comments:

CONSORT:

(2b) The objectives/hypothesis, although referenced, are not explicitly stated: (e.g., “We hypothesize that the idiom based speech training program will be more effective at xxxxx , as demonstrated by increased scores on xxx, than the conventional speech training program.”)

(7a) The authors state that the study was powered for an effect size=0.8. Could they please indicate what this means in terms of score improvement, or difference in expected scores between the two groups?

Methods

p.6. Were only patients with a first stroke included in the trial, or could patients with a history of stroke be enrolled?

The authors state that both parametric and non-parametric tests were used to compare samples as appropriate. Table 1 should reference which tests were used for each comparison, and any non-normal data should be summarized using the median and interquartile range (some of the test scores do not appear to be normally distributed).

The authors report scores of the two groups after rehabilitation treatment, but in fact, the correct comparison is between the before-and-after differences between the two groups. Therefore, the pre- and post-differences, rather than the “post” scores for each, should be reported. From the description of statistical tests used, it appears that the authors analyzed the data correctly (since both paired and unpaired tests were used). A table which clearly shows the within and between group differences would be useful to clarify this (Group 1, differences between pre- and post-intervention measurements, the same for Group 2, and the comparison between Group 1 and Group 2).

Did the authors consider doing a repeated measures analysis so that they could adjust for co-morbidities, prior strokes (if allowed), age, etc.?

Results

Please write out BA and TCM before using abbreviations.

Again, it is not clear to me why a per protocol analysis was done rather than an intent-to-treat analysis. The authors need to explain and justify their reason for so doing.

Discussion

The first paragraph of the Discussion belongs in the Introduction and should be incorporated there (some of it is redundant). The Discussion section is the place to discuss the meaning of the results, not outline the reasons for the study itself.

Second paragraph: I believe that the authors mean that the CHANGE in scores was higher in Group 1 than in Group 2. This should be clarified throughout the paper. It is never clear what the percentage changes in the two groups were, or how big of a difference there was between them, since only post-intervention scores are reported.

Fourth paragraph. The second sentence has the following note: “[Error! Reference source not found.” This implies that that the paper was not carefully proofread before submission. The actual references for these “studies” need to be added.

Fifth paragraph: The authors should refrain from using the words “we…proved…” Their study supports an association between their method and improved language skills after stroke, but does not prove it.

6. PLOS authors have the option to publish the peer review history of their article (what does this mean?). If published, this will include your full peer review and any attached files.

Reviewer #1: No

---

## [Author Response · Author response to Decision Letter 0]

16 Sep 2022

Dear reviewer and editor

Thank you for giving us the opportunity to submit a revised draft of this manuscript.We appreciate the time and effort that you and the reviewers dedicated to providing feedback on our manuscript and are grateful for the insightful comments on and valuable improvements to our paper.We have incorporated most of the suggestions made by the reviewers. Those changes are highlighted in the manuscript. Please see manuscript, in the blue (name：Super pei) , for a point-by-point response to the reviewers’ comments and concerns. In addition,we tried our best to improve the manuscript and made some changes and language editing to the manuscript. These changes will not influence the content and framework of the paper. And here we did not list the changes but marked in red ( name:Author) in the revised paper. We appreciate for Editors/Reviewers’ warm work earnestly and hope that the correction will meet with approval.

At the requirements of the editor, we have made the following revisions:

1.we have provided the ORCID iD(0000-0003-1399-4043) of the corresponding author

2.We uploaded our data at figshare.com and explained the relevant DOIs in the data availability statemen

DOI: 10.6084/m9.figshare.20766493； 

URL: https://figshare.com/articles/dataset/minimal_underlying_data_xlsx/20766493

3.We added a description of our funders to our financial disclosure，The funders had no role in study design, data collection and analysis, decision to publish, or preparation of the manuscript.

4.We have added supporting information at the end of the manuscript

5.The agency logo in the uploaded original agreement was removed

6.The file naming mode is modified

At the requirements of the reviewer, we have made the following revisions:

1. Does it explain why PP analysis is used

2. Explains why effect size=0.8.

3. The description of the patient's first stroke in the inclusion criteria was clarified

4. Modified the incorrect expression in Table 1 and added an explanation of the data tests in Table 1 Legend

5. More explicit and detailed evaluation data are presented in the results

6 adds the full name of the abbreviation

7 we changed the relevant paragraph to the introduction section and deleted the duplicate part.

8. Standardized the way of writing the manuscript

Please see the attached 'Response to Reviewers' for more specific reply information

---

## [Editor Report · Decision Letter 1]

3 Nov 2022

PONE-D-22-14223R1Effect of an extension speech training program based on Chinese idioms in patients with post-stroke non-fluent aphasia: A randomized controlled trialPLOS ONE

Dear Dr. He Xiaojun

Thank you for submitting your manuscript to PLOS ONE. After careful consideration, we feel that it has merit but does not fully meet PLOS ONE’s publication criteria as it currently stands. Therefore, we invite you to submit a revised version of the manuscript that addresses the points raised during the review process.

Some issues that highlight by the reviewers are not answered yet. Do the authors make an adjustment of the other confounding factors ? education level, side of weakness, type of treatment ? Why do you use either PP and intention to treat rather than only ITT analysis ? Please submit your revised manuscript by 12November 2022. If you will need more time than this to complete your revisions, please reply to this message or contact the journal office at plosone@plos.org. Please include the following items when submitting your revised manuscript:A rebuttal letter that responds to each point raised by the academic editor and reviewer(s). You should upload this letter as a separate file labeled 'Response to Reviewers'.A marked-up copy of your manuscript that highlights changes made to the original version. You should upload this as a separate file labeled 'Revised Manuscript with Track Changes'.An unmarked version of your revised paper without tracked changes. You should upload this as a separate file labeled 'Manuscript'.

We look forward to receiving your revised manuscript.

Kind regards,

Rizaldy Taslim Pinzon

Academic Editor

PLOS ONE

Additional Editor Comments:

Thank you for your prompt responses. Do the authors make an adjustment of the other confounding factors ? education level, side of weakness, type of treatment ? Why do you use either PP and intention to treat rather than only ITT analysis ? Please do not use point bullets for the eligibility of your subjects.
---

## [Author Response · Author response to Decision Letter 1]

6 Nov 2022

Thank you again for your letter and the reviewers’comments concerning our manuscript entitled “Effect of an extension speech training program based on Chinese idioms in patients with post-stroke non-fluent aphasia: A randomized controlled trial ”. We are very sorry that we did not explain clearly the suggestions and problems given by the reviewer in the first revision. We have read through comments carefully and have made more corrections. Based on the instructions provided in your letter, we uploaded the file of the revised manuscript. Those changes are highlighted in the manuscript. Please see manuscript, in the blue.

At the requirements of the editor, we have made the following revisions:

1.We have removed the point bullets in the Inclusion criteria and Exclusion criteria and changed them to such formats as (a)(b)(c), etc

2.We updated the manuscript and Flow diagram to intent-to-treat analysis for the data , and the missing data were filled in with baseline measurement data.

3.We conducted stratified analysis according to stroke type, education level, aphasia type and hemiplegic side.The results of our stratified analysis are included in the Response to Reviewers.

I hope that the changes I’ve made resolve all your concerns about the article. I’m more than happy to make any further changes that will improve the paper and/or facilitate successful publication.

---

## [Editor Report · Decision Letter 2]

29 Nov 2022

PONE-D-22-14223R2Effect of an extension speech training program based on Chinese idioms in patients with post-stroke non-fluent aphasia: A randomized controlled trialPLOS ONE

Dear Dr. He Xiaojun

Thank you for submitting your manuscript to PLOS ONE. After careful consideration, we feel that it has merit but does not fully meet PLOS ONE’s publication criteria as it currently stands. Therefore, we invite you to submit a revised version of the manuscript that addresses the points raised during the review process.

Thank you for making some revision.Please begin the discussion with the meaning of the results, not outline the reasons for the study itself.If you do intention to treat analysis, please begin with the flowchart. Please remove the bullet for the intervention section.Please complete the p value, in the intra group analysis. 

Second paragraph: I believe that the authors mean that the CHANGE in scores was higher in Group 1 than in Group 2. This should be clarified throughout the paper. It is never clear what the percentage changes in the two groups were, or how big of a difference there was between them, since only post-intervention scores are reported. Please submit your revised manuscript by 10 December 2022. If you will need more time than this to complete your revisions, please reply to this message or contact the journal office at plosone@plos.org. Please include the following items when submitting your revised manuscript:A rebuttal letter that responds to each point raised by the academic editor and reviewer(s). You should upload this letter as a separate file labeled 'Response to Reviewers'.A marked-up copy of your manuscript that highlights changes made to the original version. You should upload this as a separate file labeled 'Revised Manuscript with Track Changes'.An unmarked version of your revised paper without tracked changes. You should upload this as a separate file labeled 'Manuscript'.

We look forward to receiving your revised manuscript.

Kind regards,

Rizaldy Taslim Pinzon

Academic Editor

PLOS ONE
---

## [Author Response · Author response to Decision Letter 2]

1 Dec 2022

We sincerely thank the editor and all reviewers for their valuable feedback that we have used to improve the quality of our manuscript. The reviewer comments are laid out below in italicized font and specific concerns have been numbered.

1.Please begin the discussion with the meaning of the results, not outline the reasons for the study itself.

Thank you very much for your suggestion. We have revised the beginning of the discussion section,and the outline of the study at the beginning of the discussion was deleted.

2.If you do intention to treat analysis, please begin with the flowchart. 

Thank you very much for your prompt, we decided to use ITT analysis, and under your reminder, we have revised relevant parts of the paper, including the summary, description of data analysis, flow chart, and result part.

3.Please remove the bullet for the intervention section.

We are sorry that we made a mistake in the format, we have removed the bullet for the intervention section

4.Please complete the p value, in the intra group analysis. 

Thank you very much for your suggestion. In Table 2 and Table 3, we have added more accurate p values（Change from p value =0** to p value< 0.001*)

5.Second paragraph: I believe that the authors mean that the CHANGE in scores was higher in Group 1 than in Group 2. This should be clarified throughout the paper. It is never clear what the percentage changes in the two groups were, or how big of a difference there was between them, since only post-intervention scores are reported.

We are really sorry that we didn't explain it clearly here due to our inappropriate language. Under your suggestion, We have supplemented pre-intervention scores of the group I and the group II in Table 2 and Table 3. Meanwhile, we also added the percentage increase of the scores of the two groups in the first paragraph of the discussion.We would like to thank you for your suggestions to make the description of our results clearer.

We hope that the changes we’ve made resolve all your concerns about the article. We are more than happy to make any further changes that will improve the paper and/or facilitate successful publication.Thanks very much for your attention to our paprer.

---

## [Editor Report · Decision Letter 3]

23 Jan 2023

Effect of an extension speech training program based on Chinese idioms in patients with post-stroke non-fluent aphasia: A randomized controlled trial

PONE-D-22-14223R3

Dear Dr. Xiaojun,

We’re pleased to inform you that your manuscript has been judged scientifically suitable for publication and will be formally accepted for publication once it meets all outstanding technical requirements.

Kind regards,

Jan Christopher Cwik, Ph.D.

Academic Editor

PLOS ONE

Additional Editor Comments (optional):

I thank all reviewers for their time and effort in reading this manuscript. Also, thank you for your engagement during the successful revision process.
---

## [Editor Report · Acceptance letter]

27 Jan 2023

PONE-D-22-14223R3 

Effect of an extension speech training program based on Chinese idioms in patients with post-stroke non-fluent aphasia: A randomized controlled trial 

Dear Dr. Xiaojun:

I'm pleased to inform you that your manuscript has been deemed suitable for publication in PLOS ONE. Congratulations! Your manuscript is now with our production department. 

Kind regards, 

on behalf of

Dr. Jan Christopher Cwik 

Academic Editor

PLOS ONE